# Comparative Transcriptomics of Multi-Stress Responses in *Pachycladon cheesemanii* and *Arabidopsis thaliana*

**DOI:** 10.3390/ijms241411323

**Published:** 2023-07-11

**Authors:** Yanni Dong, Saurabh Gupta, Jason J. Wargent, Joanna Putterill, Richard C. Macknight, Tsanko S. Gechev, Bernd Mueller-Roeber, Paul P. Dijkwel

**Affiliations:** 1School of Natural Sciences, Massey University, Tennent Drive, Palmerston North 4474, New Zealand; elva20051202@gmail.com; 2Department Molecular Biology, Institute of Biochemistry and Biology, University of Potsdam, Karl-Liebknecht-Straße 24–25, Haus 20, 14476 Potsdam, Germany; 3School of Agriculture & Environment, Massey University, Palmerston North 4442, New Zealand; j.wargent@massey.ac.nz; 4School of Biological Sciences, University of Auckland, Auckland 1142, New Zealand; j.putterill@auckland.ac.nz; 5Biochemistry Department, School of Biomedical Sciences, University of Otago, Dunedin 9016, New Zealand; richard.macknight@otago.ac.nz; 6Center of Plant Systems Biology and Biotechnology (CPSBB), 139 Ruski Blvd., 4000 Plovdiv, Bulgaria; gechev@cpsbb.eu; 7Department of Plant Physiology and Plant Molecular Biology, University of Plovdiv, 24 Tsar Assen Str., 4000 Plovdiv, Bulgaria; 8Max Planck Institute of Molecular Plant Physiology, Am Mühlenberg 1, 14476 Potsdam, Germany

**Keywords:** multi-stress responses, Arabidopsis, comparative transcriptomics, Pachycladon, cross-species comparison, network analysis

## Abstract

The environment is seldom optimal for plant growth and changes in abiotic and biotic signals, including temperature, water availability, radiation and pests, induce plant responses to optimise survival. The New Zealand native plant species and close relative to *Arabidopsis thaliana*, *Pachycladon cheesemanii,* grows under environmental conditions that are unsustainable for many plant species. Here, we compare the responses of both species to different stressors (low temperature, salt and UV-B radiation) to help understand how *P. cheesemanii* can grow in such harsh environments. The stress transcriptomes were determined and comparative transcriptome and network analyses discovered similar and unique responses within species, and between the two plant species. A number of widely studied plant stress processes were highly conserved in *A. thaliana* and *P. cheesemanii*. However, in response to cold stress, Gene Ontology terms related to glycosinolate metabolism were only enriched in *P. cheesemanii*. Salt stress was associated with alteration of the cuticle and proline biosynthesis in *A. thaliana* and *P. cheesemanii*, respectively. Anthocyanin production may be a more important strategy to contribute to the UV-B radiation tolerance in *P. cheesemanii*. These results allowed us to define broad stress response pathways in *A. thaliana* and *P. cheesemanii* and suggested that regulation of glycosinolate, proline and anthocyanin metabolism are strategies that help mitigate environmental stress.

## 1. Introduction

Plants survive various environmental stresses by responding at physiological, biochemical, cellular and molecular levels [1,2,3,4]. Plants adapt to their environment throughout their lifetime. In addition, over evolutionary time, plants evolved to mount effective responses to environmental cues such as drought, salinity, low or high temperature, and UV-B radiation [5,6,7].

The ability to determine the expression of thousands of genes from a single sample has allowed the identification of complex stress-induced expression networks and considerable overlap between distinct stresses, and interactions within stress networks were found. A cDNA microarray analysis of cold, drought and high salinity-treated rice plants showed a wide range of cross talk involving 15 common stress-induced genes [8]. Likewise, in potato plants, 232 clones were shared between responses to cold, heat and salt stresses, as analysis of cDNA microarray data uncovered [9]. More recently, a meta-analysis of publicly accessible rice RNA-seq data revealed that photosynthesis is downregulated in response to both abiotic and biotic stress, and significant expression changes were found in genes involved in abscisic acid (ABA), jasmonic acid (JA) and salicylic acid (SA) signalling pathways [10]. In addition, this study found that several genes were uniquely induced by abiotic or biotic stress, but this was not further analysed. In *Arabidopsis thaliana*, 390 microarray samples (from 29 microarray studies) were analysed to compare drought and cold stress responses. This identified 2890 differentially expressed genes in both stress responses with similar expression patterns. Moreover, 21 drought and 16 highly inter-correlated cold gene modules were identified with four consensus gene modules, but few stress-specific genes were mentioned [11]. Osmotic stress is a primary effect caused by both drought and salt stress, while salt stress has ionic effects on cells as well [12]. Also, these stresses induce oxidative stress and damage to the membrane system [13]. Thus, drought and salt stresses have unique and overlapping responses [14]. The comparative transcriptome analysis of *Iris lactea* var. *chinensis* under drought and salt stress showed that, against the untreated control, 3897 differentially expressed genes had the same expression pattern. Nevertheless, Gene Ontology (GO) enrichment analysis revealed that while the responses were largely the same, a number of unique GO terms were enriched only in the drought or salt stress response, suggesting that this plant species can distinguish between the two related stresses [15]. Therefore, comprehensive crosstalk among multiple stress responses could be identified by high-throughput expression profiling. Yet, few studies have compared transcriptome differences between multiple stress responses of related plant species. As part of a gene co-expression analysis of four cotton species in response to cold and salt stress, 29 co-expression modules displayed significant correlations. However, four major modules were most strongly correlated with each of the four genomes, suggesting these highly co-expressed genes were species specific. This study evidenced the presence of evolutionary divergence and the potential shared network of cold and salt response among four related species [16].

In contrast with research on drought, temperature and biotic stress, research on plants’ responses to UV-B stress has been performed relatively independently to other stresses [17,18,19,20]. Nevertheless, high-throughput analyses have resulted in greater knowledge of plant UV-B radiation response pathways. Maize (*Zea mays*) plants were exposed to 9 kJ m^−2^ d^−1^ UV-B radiation, and several transcripts were found to be substantially upregulated or downregulated in one or more organs [21]. Grape (*Vitis vinifera*) leaves were subjected to UV-B radiation (4.75 kJ m^−2^ d^−1^) and microarray analysis of their gene expression responses led to classification of plant stress responses as either general protective (synthesis of UV-B absorbing compounds), antioxidant defence, pathogen defence, or abiotic [22]. RNA-seq analysis of *Lycium ruthenicum* plants subjected to UV-B radiation found 1913 upregulated and 536 downregulated genes compared to non-treated control plants, which included antioxidant-related, secondary metabolite synthesis-related, and defence response-related genes [23]. However, limited research has been conducted on the relation between UV-B radiation and other stress responses. The AtGenExpress project used Affymetrix ATH1 microarray analyses to compare heat, cold, drought, salt, high osmolarity, UV-B radiation and wounding stress regulatory networks. It was proposed that a core group of genes present in *Arabidopsis* is primarily responsible for the response to environmental stresses [24].

Here, we studied *Pachycladon cheesemanii*, a tetraploid species of Brassicaceae native to New Zealand, and compared it to its relative, *Arabidopsis thaliana*. *P. cheesemanii* can be found over an extensive latitudinal and altitudinal range, from near sea level to 1600 m altitude, and it is exposed to relatively intense UV-B radiation and winter temperatures of 2–5 °C [7,25]. Additionally, varied rock substrate (greywacke, haast schist, and basaltic and andesitic volcanic rocks) on which *P. cheesemanii* lives could impact growth via soil nutrient availability [25,26]. Thus, these diverse environmental conditions have likely caused the development of adaptive characteristics in this species, exemplified by its high UV-B tolerance [7]. In comparison, *Arabidopsis thaliana* ecotypes distribute through Western Eurasia with wide temperature ranges of −20 to +17.5 °C and precipitation (11.5–190 mm) over the year [27]. Nevertheless, the widely used laboratory accession, Col-0, was collected from Columbia, United States of America, with a daily temperature 15–16 (night)/21–22 °C (day) and a monthly precipitation of 30–70 mm (ABRC, Arabidopsis Biological Resource Center). Although the two plant species are close relatives, *P. cheesemanii* experienced a genome duplication event, a relatively short 1.61 million years ago, where one of the *P. cheesemanii* sub-genomes was proposed to have the same origin as *A. thaliana*’s genome [28]. The different living niches most likely have caused the development of species-specific characteristics over evolutionary time, while the genome duplication event may have provided the means for genes to evolve new functions in *P. cheesemanii*. Thus, we hypothesised that *P. cheesemanii* has evolved different responses to abiotic stresses from *A. thaliana*, and comparing how *P. cheesemanii* and *A. thaliana* respond to different stresses could help deepen our understanding of how the environment shapes plant evolution and adaptation.

Here, we present an interaction model describing the correlating gene networks that respond to cold, salt and UV-B radiation stress in two related plant species: *A. thaliana* and *P. cheesemanii*. The study confirmed that conserved pathways exist, but also considerable species-specific response pathways were identified, possibly driven by evolutionary selective processes.

## 2. Results

### 2.1. Abiotic Stress Transcriptomes of A. thaliana and P. cheesemanii

Changes in gene expression caused by different abiotic stresses were investigated by analysing the transcriptomes of stressed and unstressed *A. thaliana* and *P. cheesemanii* plants using an RNA-seq approach. Six-week-old *A. thaliana* and nine-week-old *P. cheesemanii* plants were treated at 4 °C, 250 mM NaCl or 5.2 μmol m^−2^ s^−1^ UV-B radiation. This UV-B radiation level was shown to cause different responses of *A. thaliana* and *P. cheesemanii* [7]. Treated and control plants were collected after 5 h of treatment to detect early transcriptional changes. Total RNA was extracted from treated and untreated leaves to result in 12 samples for each of the species; 3 biological replicates for each of the stresses and the control. The 24 RNA samples were used for RNA-seq as described in Materials and methods.

The *A. thaliana* transcriptome is available online (GenBank CP002684–CP002688) and represents 27,655 genes. We generated a *P. cheesemanii* transcriptome via RNA-seq as follows: The 12 *P. cheesemanii* Illumina libraries of 2 × 150-bp paired-end (PE) reads were sequenced for a total of 122.11 GB of sequence from 437,025,992 clean reads (Appendix A). A *de novo* transcriptome assembly of 67,905 transcripts (including alternative splicing variants) and 45,911 genes were generated as described in the Materials and Methods section (Appendix A). Transcript length distribution analysis revealed that 73.98% of transcripts were between 200 and 2000 bp, with 1% being longer than 5000 bp (Appendix A). The Benchmarking Universal Single-Copy Ortholog (BUSCO) assessment results showed that 94.7% of BUSCOs could be found in the set as single (48.7%) or duplicate (46.0%) copy. Fragmented and missing BUSCOs were rare, i.e., 2.7% and 2.6%, respectively (Appendix A). These results demonstrated high-quality integrity of the *P. cheesemanii* transcript set and validated it for use in downstream analysis.

### 2.2. Differentially Expressed Gene Number Bias in the Three Stress Responses of A. thaliana and P. cheesemanii

Using the transcriptomes of the two plant species, RNA-seq data were used to discover genes significantly differentially expressed (DE) in response to cold, salt or UV-B treatment. We used edgeR with Trimmed Mean of M-values (TMM) normalisation in all treatments [29]. Table 1 shows that between ~800 and 4000 genes were significantly DE in response to cold, salt or UV-B treatment, out of 27,655 *A. thaliana* and 45,911 *P. cheesemanii* genes. Salt stress induced the lowest number of DE genes in both plant species. A volcano plot analysis showed considerable DE gene number bias in both plant species (Figure 1), with salt stress showing ~4-fold more upregulated than downregulated genes. In *P. cheesemanii*, UV-B radiation (~4-fold) and cold (~2-fold) stress also caused a similar bias which was not found in *A. thaliana*.

Thus, while the number of DE genes in response to each stress was broadly similar in the two species, there was considerable DE gene expression bias.

### 2.3. Comparative Analysis of Stress-Responsive Gene Sets

Next, the relationship between stress responses was examined, which identified responsive genes shared among stresses, and also responsive genes that are unique to each stress within species. A comparison was made of responsive genes that were upregulated and downregulated between stresses using a Venn diagram analysis, as shown in Figure 2. As expected, upregulated transcripts due to one stress were more likely to be upregulated by another stress than downregulated, apart from one notable exception, where in *P. cheesemanii* more salt-downregulated genes were upregulated than downregulated by cold stress. Furthermore, in both plant species, salt- and UV-B-radiation-upregulated genes showed considerable overlap with cold-upregulated genes, while UV-B-radiation-upregulated genes showed little overlap with salt-upregulated ones. In general, while different stress responses shared a number of responsive genes, the majority of responsive genes were specific to each stress response within species.

Nevertheless, DE genes were selected using strict cut-off parameters and genes that were strongly upregulated under one stress may still be upregulated by another stress, but not identified as a result of the parameters used. Therefore, we also compared the stress responses by means of overlapping GO terms associated with the DE genes. GO enrichment analysis was completed and overrepresented terms for GO biological processes compared among stresses using a Venn diagram (Figure 3 and Appendix A). The overlap in GO terms was considerable between all stresses. Nevertheless, the UV-B radiation response in *A. thaliana* appeared more distinct from the others and that in *P. cheesemanii*, while the salt response included genes with comparatively less unique GO terms than those in the cold and UV-B radiation responses.

### 2.4. Network Analysis Identifies Multiple Stress-Responsive Crosstalks in A. thaliana and P. cheesemanii

To further unravel crosstalk comprising multiple stress responses, the stress transcriptomes of *A. thaliana* and *P. cheesemanii* were used to generate GO enrichment networks. GO terms were arranged in a network where connections were shaped, based on overlap between the gene sets using weighted correlation network analysis (WGCNA) and gene set enrichment analysis (GSEA), as described in the Materials and Methods section. The network layout then grouped related GO terms into network clusters to identify major overrepresented functional themes. Figure 4 shows the *A. thaliana* network analysis of upregulated pathways. Salt and cold responses shared ‘Response to stress and hormone stimulus’, ‘Response to hormone and ABA’; cold and UV-B stress responses shared ’Flavone and flavone biosynthesis’ and ‘Biological regulation’; salt and UV-B responses only shared ‘Monosaccharide signalling: glucose and hexose’. Only a few network nodes within ‘Response to UV’ were shared by all three stress responses.

For both cold and salt treatments, the downregulation network included over-represented GO terms in ‘DNA geometric change’, while cold and UV-B stress downregulated GO terms included genes involved in ‘Immune response’ and ‘Response to wounding’ (Appendix A).

Unlike the multi-stress network in *A. thaliana*, the *P. cheesemanii*’s network showed few clusters (Figure 5). ‘Response to stress’ was identified as the main cluster that included all three stress responses, which comprised some typical stress-related biological processes like anthocyanin biosynthesis and response to stress hormones.

Overall, a wide overlap was present between stress responses in *A. thaliana* in both upregulation and downregulation, while *P. cheesemanii* differed by having only one main cluster, ‘Response to stress’, identified.

### 2.5. Identification of Biological Processes Shared between A. thaliana and P. cheesemanii Stress Responses

To pinpoint biological processes shared between the two species in response to stress, overrepresented terms for GO biological processes in each stress were compared. Figure 6 shows the results of this analysis using a Venn diagram revealing that salt and UV-B stress causes overrepresentation of about the same number of unique and shared GO terms in *A. thaliana* and *P. cheesemanii*. In contrast, cold stress causes overrepresentation of many more GO terms in *P. cheesemanii* than in *A. thaliana*, and those of *P. cheesemanii* included most of those of *A. thaliana*.

Next, we aimed to obtain an overall structure of responses common to both plants. For both species and each stress response, overrepresented terms for GO biological processes (FDR < 0.05) were selected for further comparison. In response to cold, 74 overrepresented GO terms, with a similar percentage of induced genes, were shared by *A. thaliana* and *P. cheesemanii*, and the overrepresented GO terms were clustered into 13 groups (Figure 7a and Appendix A). Then, we used REVIGO to summarise and visualise the GO terms [30] as described in the Materials and Methods section. Figure 7a shows the resulting scatterplot of the common cold response, where the GO terms are placed in a semantic space, and clusters indicate semantic similarity among the terms within that cluster. Representative GO terms are then shown based on their dispensability values (<0.15) and visual inspection. The results suggest that in both *A. thaliana* and *P. cheesemanii,* a wound-like response is initiated and that circadian rhythm (in plants only circadian rhythm is relevant within rhythmic process) as well as secondary metabolism, including that of flavonoids, trehalose, phenylpropanoid and oxylipin, is important during the early cold response. Most of the representative GO terms had similar percentages of upregulated and downregulated genes in *A. thaliana* and *P. cheesemanii* except flavonoid metabolism and regulation of response to stimulus (Appendix A).

The common salt response included 26 GO terms, and the REVIGO analysis indicated relatively generic GO terms, including the cluster ‘Response to stimulus’, ‘Regulation of abscisic acid-activated signalling pathway’ and ‘Regulation of response to stimulus’ (Figure 7b). The results suggest that the salt response may either be a generic response, or both species may respond to this stress in unique ways (Appendix A).

The common UV-B radiation response included 65 overrepresented GO terms and REVIGO indicated clusters summarised by ‘Vitamin, L-phenylalanine, chorismite, L-ascorbic acid, and aromatic amino acid metabolic process’ and ‘Response to stimulus’. Other representative terms include those associated with secondary metabolism and pigments in particular and photosynthesis and light harvesting. The representative GO terms of these clusters also had similar rates of upregulated and downregulated genes in both species except ‘Cellular amine metabolic process’, ‘Response to stimulus’, and ‘Secondary metabolic process’ (Appendix A).

Overall, the analysis suggests that both species commonly use generic processes, including changes in secondary metabolism, as an early response to stress.

### 2.6. Identification of Unique Biological Processes in A. thaliana and P. cheesemanii Responses to Three Stresses

To achieve an overall picture of specific responses of each species, unique overrepresented terms for *A. thaliana* and *P. cheesemanii* biological processes in each stress response were selected for REVIGO analysis as described in the Materials and Methods section. Figure 8 shows the resulting six scatter plots representing the unique cold, salt and UV-B radiation response sets for *A. thaliana* and *P. cheesemanii*. The *A. thaliana* cold-response set is largely encompassed by that of *P. cheesemanii* (Figure 6), and in Figure 8a this is reflected by a few unique representative terms for *A. thaliana*: notably, ‘Jasmonic acid metabolic process’, ‘Negative regulation of ethylene-activated signalling pathway’, ‘Hyperosmotic salinity response’ and ‘Disaccharide and oligosaccharide metabolic process’. The unique *P. cheesemanii* response set is much larger and includes ‘Cell wall organisation or biogenesis’, ‘Positive regulation of cell communication’ and ‘L-phenylalanine, proline, cinnamic acid, oxylipin, and glycosinolate metabolic process’. In response to salt, both species activate a similar number of unique GO terms. Major unique *A. thaliana* clusters are ‘Regulation of response to biotic stimulus’, ‘Wax biosynthetic process’ and ‘Oxylipin, jasmonic acid metabolic process’, while the terms for *P. cheesemanii* include ‘Catabolic processes’, ‘Stomatal movement’, ‘Tetrapyrrole and proline catabolic process’ and ‘Ageing and leaf senescence’ (Figure 8b). Upon exposure to UV-B radiation, *A. thaliana* responds with ‘Photosynthesis’, ‘Response to red light’, ‘Indole glucosinolate metabolic process’, ‘Defence response by callose deposition in cell wall’, ‘Embryo development’ and several stress-related hormone metabolic processes, while unique *P. cheesemanii* responses are ‘Wound healing’, ‘Regeneration’, ‘Positive regulation of flavonoid biosynthetic process’, ‘Maltose metabolic process’ and ‘Anthocyanin-containing compound cinnamic acid, galactose, chlorophyll, and L-phenylalanine biosynthetic process’ (Figure 8c).

## 3. Discussion

Plants must respond to a wide range of abiotic and biotic environmental stresses and different plant species have developed unique strategies for dealing with these challenges [31,32,33,34]. *P. cheesemanii* is a close relative of *A. thaliana*, and its tetraploid genome may have contributed to its ability to survive in a wide range of habitats. The work presented here aimed to detect unique stress response pathways in *P. cheesemanii*. Both species showed similarity in the number of responsive genes upregulated and downregulated under different stresses, with the exception that UV-B radiation downregulated much fewer genes in *P. cheesemanii* (Table 1 and Figure 1), and in terms of overrepresented GO terms, *A. thaliana* induced more responses to this stress. In addition, *P. cheesemanii* displayed a broader response to cold stress compared to the other two stresses (Figure 3). These findings suggest that *A. thaliana* and *P. cheesemanii* induce some similar responses to each stress, but that there should also be unique stress-responsive processes in each species.

### 3.1. Classical Stress Responsive Processes Are Conserved in Both A. thaliana and P. cheesemanii

General plant stress-responses have been identified through analyses of plant stress transcriptomes and include the response to stimulus, regulation of response to stimulus, multi-organism process, biological regulation and signalling [10,35,36,37]. Unsurprisingly, these processes were found in both *A. thaliana* and *P. cheesemanii* responding to multiple stresses (Figure 7). For instance, three out of four overrepresented biological processes (multi-organism process, response to stimulus and response to endogenous stimulus) induced by salt stress in both *A. thaliana* and *P. cheesemanii* were induced by cold and UV-B stress as well. Only the regulation of the abscisic acid-activated signalling pathway was specific to salt stress in both plant species (Figure 7b).

Cold stress induced 13 common biological processes in both *A. thaliana* and *P. cheesemanii*. These included metabolic processes of trehalose, phenylpropanoid and oxylipin and rhythmic process, with the latter being highly specific to cold stress (Figure 4 and Figure 7a). The circadian rhythm is coordinated with environmental signals to maintain plant fitness and survival via various hormone pathways and contributes to regulation of seed germination, leaf growth, photosynthesis and flowering [38,39,40]. The circadian rhythm regulates abiotic stress responses in a wide range of plants, including Arabidopsis, soybean, barley and rice [41,42,43,44,45]. Key circadian clock regulators, CCA1, LHY, CHE, TIC and TOC1, regulate stress responses via crosstalk with salicylic acid, jasmonic acid and ethylene signalling pathways [46,47]. *TIMING OF CAB EXPRESSION 1* (*TOC1*) can be induced by ABA treatment and then contribute to ABA signalling induction [48]. Overexpressed *TOC1* resulted in drought hypersensitivity due to reduced stomatal closure [48]. The circadian clock furthermore regulates the extent of induction of genes encoding C-repeat Binding Factor 1/Dehydration Responsive Element Binding 1 (*CBF1*/*DREB1*) family of transcription factors, which contribute to cold tolerance [49]. The putative MYB transcription factor, cold-induced *MYB* (*CMYB1*) was found to respond to circadian rhythm in rice leaves at different developmental stages [50]. Induction of secondary metabolism is another common cold response in plants, and genes involved in trehalose, oxylipin and phenylpropanoid biosynthesis were upregulated in *P. cheesemanii* and *A. thaliana* in this study. Notably, these metabolites interact with circadian rhythm and hormone pathways [51,52,53]. Moreover, flavonoids influence the expression of circadian clock genes as found by RNA-seq analysis of an Arabidopsis flavonoid biosynthesis mutant [54]. Elevated trehalose biosynthesis increased drought, salt, cold, and heat tolerance in tobacco, potato, Arabidopsis and rice [55,56,57]. In *A. thaliana*, trehalose also interacts with JA and SA signalling after heat stress via multiprotein bridging factor 1c (MBF1c), a protein involved in controlling thermotolerance [58]. In maize, JA-deficient mutants revealed that wound-induced oxylipin responses are positively regulated by JA signalling [52]. In addition, other secondary metabolites, phenylpropanoids, were induced by salinity, drought, temperature and UV radiation stress. The downstream phenolic compounds of phenylpropanoid metabolism in turn scavenged stress-causing free radicals like ^1^O_2_, O_2_^2−^, OH^−^, thereby protecting plant membranes from stress-induced peroxidation damage [59,60].

In both *A. thaliana* and *P. cheesemanii*, UV-B radiation impacted pathways involved in metabolism of L-ascorbic acid, L-phenylalanine, and chorismate from aromatic amino acid metabolic process (Figure 7c). Besides their structural role in proteins, phenylalanine, tyrosine and tryptophan are precursors of a number of phytohormones such as auxin and SA as well as aromatic secondary metabolites in plants and micro-organisms [61]. Phenylalanine can be deaminated by phenylalanine ammonia-lyase to produce phenylpropanoid compounds like flavonoids, anthocyanins, flavonols, and flavones [62]. Flavonoids are important pigments that protect against UV-B-induced damage because of their antioxidant capacity and UV radiation absorption capacity [63]. In addition, genes involved in L-ascorbic acid (vitamin C) biosynthesis were impacted by UV-B radiation in both *A. thaliana* and *P. cheesemanii* (Figure 7c), and also in other plant species [64,65]. In cucumber seedlings, low-fluence UV-B radiation (20 μW cm^−2^) elevated L-ascorbic acid abundance and its biosynthesis genes (*CsGLDH*, *CsMIOX1*, *CsAO2*, *CsAO4*, *CsAPX5*, *CsGR1*, and *CsDHAR1*) and light-responsive elements were identified in their promoter regions [64]. Thus, biosynthesis of secondary metabolites appears a conserved strategy to cope with UV-B radiation.

In conclusion, a number of widely studied plant stress processes are highly conserved in *A. thaliana* and *P. cheesemanii* as well (Figure 9). This suggests that these responses likely function in adapting generic processes such as growth and development in many plant species in order to minimise the impact of the stresses and provide time for plants to adapt to the stress and finish their lifecycle.

### 3.2. Unique Cold, Salt, and UV-B-Radiation Responses in A. thaliana and P. cheesemanii

Plants cope with challenging temperatures by adopting different strategies, one of which is the accumulation of low molecular weight carbohydrates (LMWC) [66]. We found that cold stress induces genes involved in biosynthesis of LMWC metabolites (oligosaccharides) in *A. thaliana* uniquely (Figure 8a). Galactinol synthase (GolS) catalyses the first step of the biosynthesis of the raffinose family of oligosaccharides (RFO) and overexpression of *GolS*, increases endogenous galactinol and raffinose and improves drought tolerance in rice [67]. Its overexpression also increases the tolerance of transgenic plants to osmotic and salinity stresses, and increased levels of galactose and raffinose were found in a chilling-tolerant genotype of *Oryza sativa* after cold stress [68,69]. RFOs likely play a role in reactive oxygen species (ROS) scavenging under stress [70]. Oligosaccharides themselves can also trigger various stress-related signalling pathways [71,72,73]. Treatment with chitosan oligosaccharides elevates JA content in multiple plant species and the expression of JA-related genes in *Brassica napus* [74,75,76,77]. A study of *jar1* (JA-deficient), *NahG*, and *sid2* (SA-deficient) mutants in Arabidopsis suggested the involvement of both SA and JA signalling in chitosan oligosaccharide-induced resistance to *Pseudomonas syringae* pv. *tomato* DC3000 [78]. This could be the reason for ‘Jasmonic acid metabolic process’ and ‘Negative regulation of ethylene activated signalling pathway’ in *A. thaliana* unique responses to cold stress in this study (Figure 8a). In contrast, the responses of *P. cheesemanii* to cold stress included many more overrepresented terms for GO biological processes relative to those in responses to salt and UV-B radiation (Figure 3), implying the complexity of *P. cheesemanii* cold responses. Under cold stress, the responses to multiple plant hormones, including salicylic acid, ethylene and gibberellin, as well as a couple of stress-related metabolites have indeed been found in *P. cheesemanii* cold response (Figure 5 and Figure 8a). Another noticeable process was glucosinolate metabolism and relevant processes (glycoside catabolic process and sulfate reduction). Glucosinolates are a class of glycosinolates whose sugar component is glucose and which are primarily found in *Brassicaceae* [79]. The accumulation of glucosinolates is induced by a variety of abiotic stresses such as salinity, drought, temperature and nutritional deficiencies [80,81,82,83]. Arabidopsis TU8 mutants exhibited a deficiency in glucosinolate metabolism and displayed less tolerance to high temperatures, while exogenous application of glucosinolate derivatives strengthened the heat tolerance of *A. thaliana* plants [84,85]. However, there was little evidence to support a role for glycosinolate metabolism under low temperatures. In contrast, here, we identified glycosinolate metabolism as a unique cold response in *P. cheesemanii* (Figure 8a). It might be a stress-acclimating strategy similar to that of glucosinolate-induced heat tolerance, and this hypothesis would be interesting to follow up in further studies.

It has been reported that wax biosynthesis responds to salinity in a variety of plant species, which also has been found in this study [86,87,88]. Stress-induced wax biosynthesis has been linked to enhanced plant tolerance to abiotic stresses like low temperature and drought [89,90]. In this study, it could be associated with cuticle development in *A. thaliana*’s unique salt response, as waxes are main components of the plant cuticle [91]. The cuticle plays a role in protection against water loss and its biosynthesis is responsive to environmental stress [92]. Salt stress causes accumulation of alcohols, which are wax components, and this may stimulate sugar beet growth [93]. The *A. thaliana shine* gain-of-function mutant displayed increased and altered wax composition accompanied by increased cuticle permeability, reduced stomatal density and drought tolerance [94]. Similarly, the AP2 domain-containing putative transcription factor gene *WXP1* of *Medicago truncatula* can increase wax production and confer drought tolerance in *Medicago sativa* [95]. Alteration of the cuticle is widely reported in response to salt and drought stress, but, interestingly, GO terms related to this process were not overrepresented in *P. cheesemanii*.

Overrepresented pigment and chlorophyll, as well as tetrapyrrole and proline catabolic processes GO terms, were unique in the *P. cheesemanii* salt response (Figure 8b). It was reported that foliar application of the tetrapyrrole precursor 5-aminolevulinic acid (ALA) onto salt-stressed *Brassica napus* seedlings increased chlorophyll and proline abundance and improved tolerance to salt [96]. The concentration of the tetrapyrrole chlorophyll was reduced under salt stress, resulting from a declined leaf water potential that limited the photosynthetic rate and disrupted the biosynthesis of chlorophyll [96]. Elevated proline abundance is a salt-induced phenomenon in many plant species, including pepper, maize, melon, and sorghum, and it alleviates the effect of salinity stress in a number of ways, including inhibition of the stomatal opening [97,98,99,100,101]. Notably, the GO term ‘Stomatal movement’ was overrepresented in *P. cheesemanii* in response to salt stress (Figure 8b). Two major responses to salt and the related drought stress are alteration of the cuticle and biosynthesis of proline. Curiously, the former appears to be working only in *A. thaliana* and the latter only in *P. cheesemanii*. This suggests an important evolutionary deviation between these related species and it will be interesting to discover the functional and evolutionary basis of this difference.

A unique UV-B radiation response in *A. thaliana* was the enrichment of GO terms related to protecting actions like callose-related cell wall defence (´Polysaccharide localisation´) (Figure 8c). This may also contribute to plants’ acquired resistance to biotic stresses following UV-B radiation [102]. In *A. thaliana*, exposure to sub-lethal UV-C radiation increased the production of callose [103], and UV-B radiation induced callose ring formation and cell wall thickening of the upper part of the trichome [104]. Interestingly, GO terms related to glucosinolate metabolism were also uniquely impacted by UV-B radiation in *A. thaliana* (Figure 8c). Glucosinolate-dependent callose deposition is part of the Arabidopsis innate immune response against microbial pathogens [105]. Derivates derived from glucosinolate hydrolysis were suggested to function as insect feeding and oviposition deterrents in insect resistance; they also contribute to microbe-associated molecular pattern-mediated defence as signalling molecules [106,107]. While callose-related cell wall defence is correlated with pathogen tolerance, callose deposition was also observed in response to abiotic stress, with unknown physiological and molecular mechanisms [108]. Therefore, our observations support the existence of cross-tolerance to biotic and abiotic stresses in plants [109]. After UV-B radiation treatment of *P. cheesemanii*, the enrichment network analysis identified a unique cluster related to anthocyanin biosynthesis and included the regulation of L-phenylalanine (biosynthetic precursor of anthocyanin) and anthocyanin metabolism (Figure 8c). This observation further implied an important role for anthocyanins in *P. cheesemanii* stress responses (Figure 5). Anthocyanins are a widely distributed group of water-soluble flavonoid pigments, and their biosynthesis involves more steps than those of flavone and flavonol biosynthesis [110]. Coupled with steroid metabolism, anthocyanin accumulation was associated with reduced membrane damage in other plant species and this could help stabilise membrane systems and minimise the ROS damage caused by UV-B radiation [111] in *P. cheesemanii* (Figure 7c and Figure 8c). In conclusion, the unique UV-B radiation response in *A. thaliana* suggests that changes at the cell, organ and whole plant level help adapt the plant to UV-B radiation, while in *P. cheesemanii* anthocyanin production may be a main strategy for coping with this stress.

While *A. thaliana* and *P. cheesemanii* share common stress responses, they also display considerable differences, even though they are evolutionarily relatively closely related (Figure 9). The natural habitat of the two species is quite different [7] and our results suggest that plants evolve unique stress response pathways quickly. A better understanding of the shared and unique stress-responsive pathways of *A. thaliana* and *P. cheesemanii* could help to model common stress responses in all plant species but also provides insight into the range of potential responses that help mitigate environmental stress.

### 3.3. Prominent Brassicaceae Pathways May Contribute to a Variety of Stress Responses

In a comprehensive transcriptomic comparison of responses to cold, salt, heat and drought stress between the distantly related plants Arabidopsis and rice, there was an overlap in the abiotic stress transcriptome of the two species [112]. However, both had significant differences in responding to stresses. Only ~14% of differentially expressed genes in rice had Arabidopsis orthologues with a similar response under drought and salt stresses; the two species shared less than 25% of the orthologues that have common responses in the heat and cold-responsive sets, although more than 60% of the responsive genes in two sets had orthologues in Arabidopsis and rice. In addition, a number of orthologous genes were responding to stresses in an opposite manner. Besides, there was a large proportion of stress-responsive non-orthologous genes in rice and Arabidopsis. The downregulated orthologous under drought stress represented ~10% of downregulated genes in rice and 44% in Arabidopsis. In terms of functional representation, Arabidopsis and rice have similar over/under-representation patterns in three functional categories, while there were significant differences between the two species in drought, salt, cold and heat stress responses [112]. In contrast, the transcriptomes of the closely related Brassicaceae *A. thaliana* and *P. cheesemanii* overlapped largely in three stress response sets. For example, 127 out of 166 overrepresented GO terms in *A. thaliana* overlapped with 354 *P. cheesemanii* overrepresented GO terms in responding to cold stress (Figure 6). Additionally, the two species shared characteristics with other species from the Brassicaceae family. Plant species in this family contain high phenolic and glucosinolate levels, which may play a role in detoxifying excess ROS resulting from biotic and abiotic stresses [113,114,115]. In Brassicas, biotic or abiotic stresses increased the production of amino acids, sugars, indoles, phenolics and glucosinolates [81,115,116,117,118] and activated JA, SA, and ABA signalling pathways, resulting in the initiation of the plant defence responses [119,120,121]. In this study, amino acid and phenolics-related metabolism processes were overrepresented in *A. thaliana* and *P. cheesemanii* cold responses, and sugar, indole and glucosinolate-related metabolism processes were overrepresented in *A. thaliana* and *P. cheesemanii*, responding to different stresses (Figure 8a,c). Moreover, hormone and defence-related processes were found to be overrepresented in both species and in response to three abiotic stresses. (Figure 4 and Figure 5). These results are consistent with the idea that over evolutionary time, some stress-related metabolites and stress-response mechanisms that are prominent in *Brassicaceae* have been repurposed in *A. thaliana* and *P. cheesemanii* to respond to different stresses. In addition, the two species have evolved their unique responses to the same stress as described above. *A. thaliana* and *P. cheesemanii* are relatively closely related but the ecological niches they occupy are quite distinct [7] and this may be the reason for the divergence in their response to stresses. In response to cold, *P. cheesemanii* may employ more primary and secondary metabolites (L-phenylalanine, pro-line, cinnamic acid, oxylipin and glycosinolate) to limit cold-induced damage or activate stress signalling pathways. Such multiple strategies could help this species survive in an Alpine area with rapidly changing temperatures. Similarly, *P. cheesemanii* may have evolved a more well-developed strategy to cope with the relatively high levels of UV-B radiation in its living niches (Figure 9).

The two subgenomes of *P. cheesemanii* are likely another driving force behind the observed divergence in stress responses. The historic genome duplication could have provided the genetic basis for the development of new gene functions [122,123] and may have contributed to the redirection and diversification of the stress response pathways already present. In this study, few analyses were performed on the relationship between the polyploidy of *P. cheesemanii* and its stress responses, due to the difficulty of correctly assembling homologous transcripts. Further study on this aspect would help us to delve into the interaction of plant polyploidization and adaptation.

## 4. Materials and Methods

### 4.1. Plant Growth and Stress Treatments

Seeds of *P. cheesemanii* Kingston (geographical coordinates in decimal degrees −45.3273, 168.7078) was provided by Dr. Claudia Voelckel (Max Planck Institute for Chemical Ecology, Jena, Germany) and Dr. Peter Heenan (Wildland Consultants Ltd., Rotorua, New Zealand). Seeds of *Arabidopsis thaliana* Heynh. accession Col-0 was obtained from the Arabidopsis Biological Resource Center (ABRC; https://abrc.osu.edu). Seeds of both species were sown and germinated in wet Seed Raising Mix^®^ soil from Daltons (Matamata, New Zealand) and seedlings were grown under a 16 h light (200 μmol m^−2^s^−1^ cool white fluorescent tube)/8 h dark (long-day) regime at 22 °C. For multiple stress transcriptome profiling, seedlings were grown under a 10 h light (200 μmol m^−2^s^−1^ cool white fluorescent lamp)/14 h dark (short-day) regime at 22 °C for six weeks (*A. thaliana*) or nine weeks (*P. cheesemanii*) (Appendix A). *A. thaliana* and *P. cheesemanii* plants with different ages were used due to the distinct growth characteristics of the two species. Six-week-old *A. thaliana* and nine-week-old *P. cheesemanii* plants appeared similar [7] and were used for the experiments. For cold stress, plants were transferred to a 4 °C growth chamber with otherwise the same light settings. For salt stress, pots containing the plants were saturated with a 250 mM NaCl solution and excess solution was allowed to drain. The plants were transferred to the UV-B radiation chamber where they were subjected to normal white light (200 μmol m^−2^s^−1^ cool white fluorescent tube) supplemented with 5.2 μmol m^−2^s^−1^ UV-B (290–320 nm) for UV-B treatments, while the control plants were kept under white light conditions [7]. The UV-B fluorescent tubes used in the chamber were Q-Panel 313 (Q-Lab Corp, Cleveland, OH, USA), which were wrapped in 0.13 mm-thick cellulose diacetate foil (Clarifoil; Courtaulds Ltd., Derby, UK) to remove wavelengths < 290 nm. The chamber was split into a UV-B+ zone and a UV-B− zone separated by a central curtain of UV-B opaque film (Lumivar; BPI Visqueen, Ardeer, UK). For the UV-B− zone, the UV-B tubes were wrapped in the same UV-B opaque film [124]. UV-B treatments were quantified at plant canopy height with an Optronics OL-756 UV-VIS Spectroradiometer (Optronic Laboratories, Gooch and Housego, FL, USA) equipped with an integrating sphere. Spectroradiometric scans of the controlled environment chamber confirmed that the biologically effective UV dose was < 0.01 kJ m^−2^d^−1^ in the UV-B− zone (Appendix A). Plants were treated one hour after the lights turned on (simulating dawn); then, five hours later (6 h after dawn), the two largest, fully expanded mature leaves of treated and nontreated plants were collected.

### 4.2. Library Preparation and Illumina Transcriptome Sequencing

Total RNA was extracted from mature leaves using a Quick-RNA MiniPrep Kit (Zymo Research, Irvine, CA, USA) and treated with DNase I to remove genomic DNA contamination (Appendix A). Purified untreated and treated plant RNA was used to generate 150 bp paired-end sequencing libraries, including 12 *A. thaliana* and 12 *P. cheesemanii* samples (three biological replicates for untreated plants and each stress treatment in each plant species). Following library quality control, the libraries were sequenced on Illumina HiSeq X Ten (San Diego, CA, USA), with 2 × 150-PE reads generating a total of ~278.2 Gb raw sequencing data. PE library construction and Illumina sequencing were performed by Novogene Limited (Beijing, China).

### 4.3. Pachycladon Transcriptome Assembly

Sequencing adaptors were removed from sequenced reads using trim_galore v0.4.1 [125] and ribosomal RNA was filtered out by using SortMeRNA v2.1b [126]. The quality filtered reads were normalised in silico using Trinity v2.5.1 [127]. Since a genome draft was assembled in our previous study, the reference genome-guided transcriptome assembly was applied [7]. However, the results from Bowtie2 and BUSCO showed that the quality of the assembled transcriptome was not optimal, as shown by a low-percentage Bowtie and BUSCO alignment. Therefore, de novo transcriptome assembly was performed using three programs with the results compared to generate a high-quality transcriptome. Trinity, Velvet/Oases and Trans-ABySS have been suggested as providing better performance for de novo transcriptome assemblies [128]. The normalised read sets were then independently assembled using Trinity v2.5.1 [127], Velvet v1.1/Oases v0.2 [129,130], and Trans-ABySS v2.0.1 [131] assemblers using a range of k-mer sizes (Appendix A). For each assembler, a popular range of k-mer sizes was selected (Trinity: 19–31-mer; Trans-ABySS: 51–63-mer; Velvet/Oases: 55–95-mer). BUSCO analysis was used to confirm whether the best assembly was achieved in the selected range. BUSCO (v3.0.2; dataset: ‘embryophypta_odb9’, containing 1440 orthogroups, downloaded from http://busco.ezlab.org, access on 1 March 2019) [132] and the percentage of reads aligned using Bowtie2 were used as assembly quality metrics to select the Trans-ABySS assembly based on its superior completeness and accuracy. The resulting 19 assemblies were evaluated using the Bowtie alignment rate and near-universal orthologue searching (Appendix A). Seven assemblies produced by Trans-ABySS generated higher read alignment rates (~91.08%) than those from the other two assemblers (~87.23% from Velvet/Oases and ~88.91% from Trinity). The BUSCO results generally showed high percentages of complete BUSCO across all assemblies (Appendix A), with the exception of the 19-mer assembly from Trinity, which had higher percentages of fragmented and missing BUSCOs.

As the assemblies from the Trans-ABySS assembler outperformed those of Velvet/Oases and Trinity, based on a Bowtie evaluation, they were selected for further processing. Next, Trans-ABySS assemblies across different k-mer sizes were combined to generate the final transcript set (318,111 transcripts), which was further clustered and assembled using the CAP3 [133] assembly program. This program removed technical redundancy with 99% overlap in percentage identity and 200 bp overlap in length, resulting in 223,341 transcripts (Appendix A). Finally, EvidentialGene: tr2aacds [134] was applied to remove redundancies and fragments and to identify transcript splice isoforms.

### 4.4. Functional Annotation of The Pachycladon Transcriptome

Homology-based annotation was performed using BLASTP v2.6.0 [135] against the Uniprot *A. thaliana* dataset (Swissprot + TrEMBL) using the parameters best hit, E-value cut-off of 1 × 10^−20^, query coverage of ≥50% and percentage identity ≥50%. GO annotations were obtained from the Uniprot database. To annotate the final transcript set, the sequences were searched against the *A. thaliana* UniProt database using BLASTX with an E-value cut-off of 1 × 10^−5^. Of the 45,911 genes, 39,949 had homologies in the UniProt database with >50% identity. These genes were mapped to 29,060 *Arabidopsis* proteins with 5294 GO annotations. The overrepresented GO terms for biological processes were ‘Carbohydrate metabolic process’ (735 members), ‘Cell redox homeostasis’ (352 members), ‘Cell wall organisation’ (365 members), ‘Defence response’ (430 members), ‘DNA integration’ (493 members), ‘Intracellular protein transport’ (487 members), ‘Protein transport’ (472 members), ‘Regulation of transcription, DNA-templated’ (1418 members), ‘Signal transduction’ (415 members) and ‘Translation’ (960 members) (Appendix A). There were 78% transcripts which could be annotated against the *A. thaliana* protein database, as expected, suggesting that the assembled transcriptome was suitable for downstream analysis.

### 4.5. Analysis of Differential Gene Expression

To analyse differential gene expression induced by stress, the 147.1 Gb raw data with 490,400,436 raw reads from the *A. thaliana* stress response sequencing libraries was processed using trim_galore [125] to remove adaptor contamination. SortMeRNA [126] was then used to remove ribosomal RNA sequences from the adaptor-trimmed reads. Then, *A. thaliana* and *P. cheesemanii* reads from untreated and treated samples were mapped to the *A. thaliana* transcriptome reference (GenBank CP002684–CP002688) and the de novo assembled *P. cheesemanii* transcriptome using kallisto v0.43.1 [136], respectively. The *A. thaliana* reference used to quantify transcripts was downloaded from www.araport.org (v10). The parameters of kallisto/edgeR analysis were: counts per million (cpm) > 1 (removing low count), false discovery rate (FDR) < 0.05 and log of fold change (FC) ≥ 1. Differential gene expression was then performed using edgeR v3.26.1 [137] with default parameters.

### 4.6. Identification of Shared and Unique Biological Processes of Species’ Stress Responses

A number of biological processes were identified as being shared or not shared between *A. thaliana* and *P. cheesemanii* responding to each stress. To obtain an overall structure of responses common to the species for each stress, the overrepresented terms in common between the *A. thaliana* and *P. cheesemanii* biological process in each stress response, with a 0.05 adjusted *p*-value (adjusted by Benjamini and Hochberg correction), were selected as significantly overrepresented terms. This resulted in three sets: a common cold response set; a common salt response set; and a common UV-B radiation response set. Because of the well-annotated *A. thaliana* transcriptome, *A. thaliana* genes from these three sets were used to generate three gene subsets for further GO ontology and clustering analysis. Each subset was scrutinised to identify clusters of overrepresented terms for GO biological processes and then the identified clusters were annotated functionally.

To achieve an overall picture of each species’ unique response to each stress, unique overrepresented terms for *A. thaliana* and *P. cheesemanii* biological processes in each stress response (with a 0.05 adjusted *p*-value) were selected as significantly overrepresented terms. This resulted in six sets: one unique cold, salt and UV-B radiation response set for both *A. thaliana* and *P. cheesemanii*. All genes were extracted from these six sets to generate six gene subsets. Each subset was scrutinised to identify clusters of overrepresented terms for GO biological processes and then the identified clusters were annotated functionally.

### 4.7. Combining Weighted Correlation Network Analysis and Gene Set Enrichment Analysis

Weighted correlation network analysis (WGCNA)-based gene modules were used as gene sets for gene set enrichment analysis (GSEA). Twelve datasets from the three stress responses (three biological replicates) in *A. thaliana* were utilised for WGCNA. The stress-responsive genes could be moulded into 11 groups; up- and downregulated genes were identified in each. GSEA v4.0.3 [138] was performed on 11 WGCNA modules in each stress and the network was then generated and visualised using EnrichmentMap and the compound spring embedder (CoSE) layout in Cytoscape v3.8.0 [139]. For the interaction network analysis of stress responses in *P. cheesemanii*, GSEA was applied to the datasets. The resulting network was also generated and visualised using EnrichmentMap and the CoSE layout in Cytoscape.

## Figures and Tables

**Figure 1 ijms-24-11323-f001:**
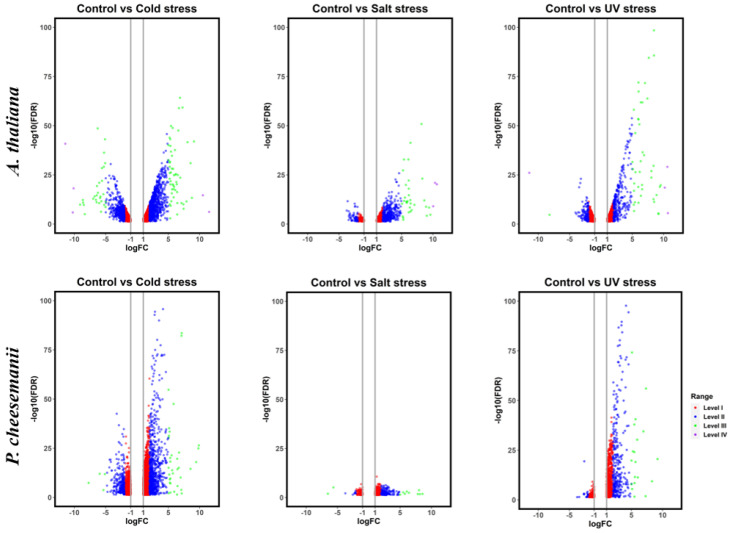
Differential gene expression of *P. cheesemanii* and *A. thaliana* caused by different stresses. Volcano plots, showing fold change in abundance (logFC) against significance, indicate DE genes in response to cold, salt and UV-B radiation stress. FC: fold change; FDR: false discovery rate. Upper row: *A. thaliana*; lower row: *P. cheesemanii*. The differentially expressed genes were grouped into 4 levels with different colours: Level I (orange), with 2 ≤ |FC| < 4; Level II (blue), with 4 ≤ |FC| < 32; Level III (green), with 32 ≤ |FC| < 1024 and Level IV (purple), with 1024 ≤ |FC|. FC, fold change.

**Figure 2 ijms-24-11323-f002:**
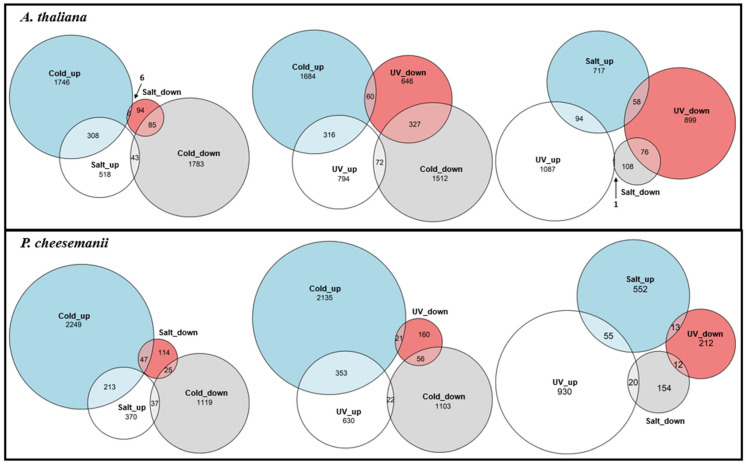
Venn diagrams of up- and downregulated genes in response to stress in *A. thaliana* and *P. cheesemanii*. The diagrams show the number of up- and downregulated genes and overlap in response to cold, salt, and UV-B radiation. Upper row: *A. thaliana*; lower row: *P. cheesemanii*.

**Figure 3 ijms-24-11323-f003:**
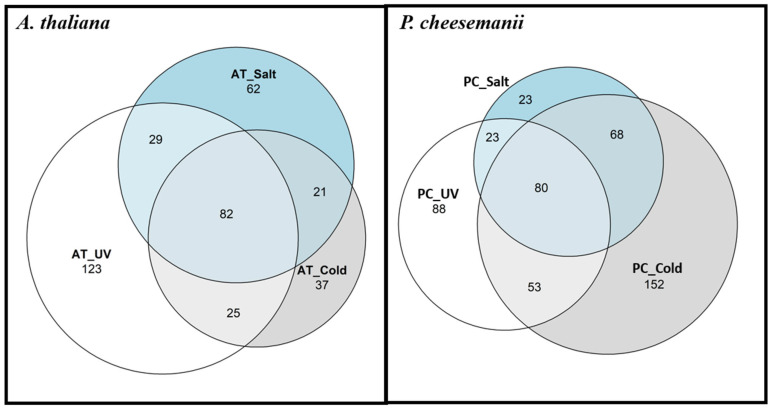
Venn diagram of the numbers of GO terms of biological processes between stresses in *A. thaliana* and *P. cheesemanii*. AT: *A. thaliana*; PC: *P. cheesemanii*. UV circles: the number of overrepresented GO terms in UV-B radiation response; Cold circles: the number of overrepresented GO terms in cold response; Salt circles: the number of overrepresented GO terms in salt response.

**Figure 4 ijms-24-11323-f004:**
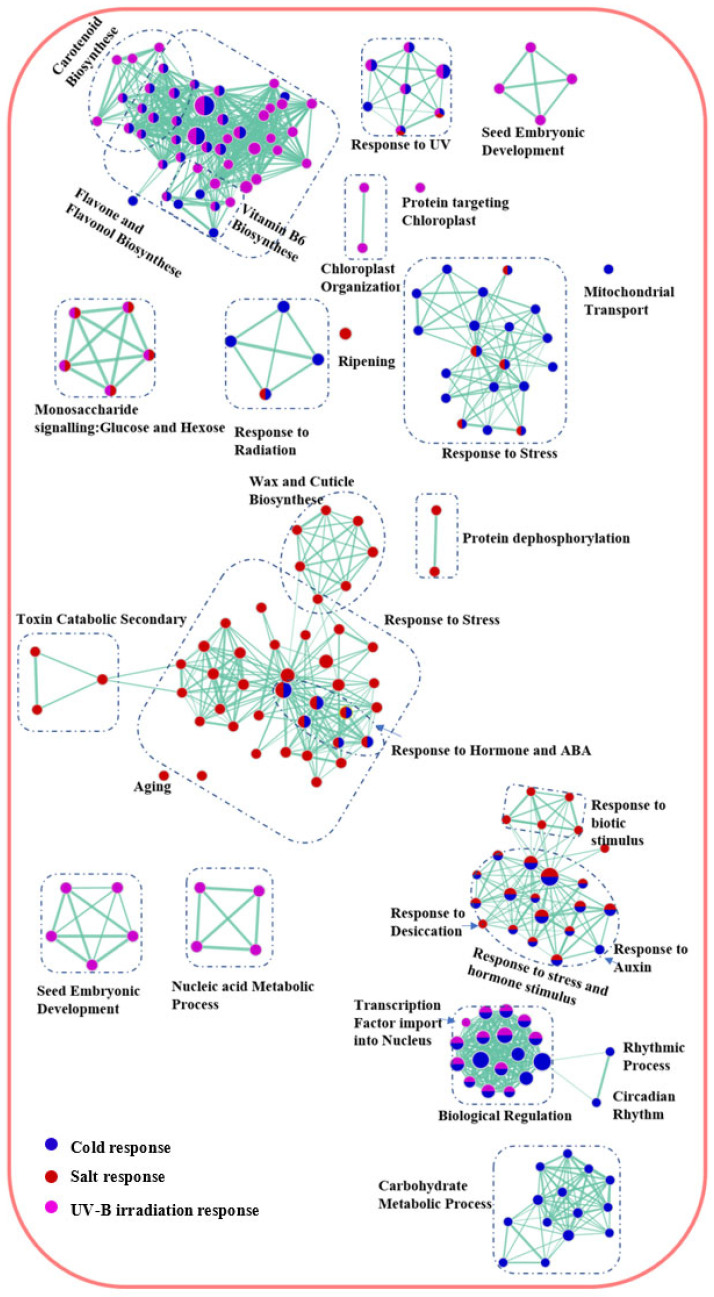
Network analysis of upregulated biological processes of A. thaliana multiple stress-responsive transcriptomes. Purple, red and blue circles: overrepresented GO terms in response to UV-B radiation, salt and cold stress, respectively. Boxes with dashed lines: clusters of overrepresented GO terms involved in similar biological processes.

**Figure 5 ijms-24-11323-f005:**
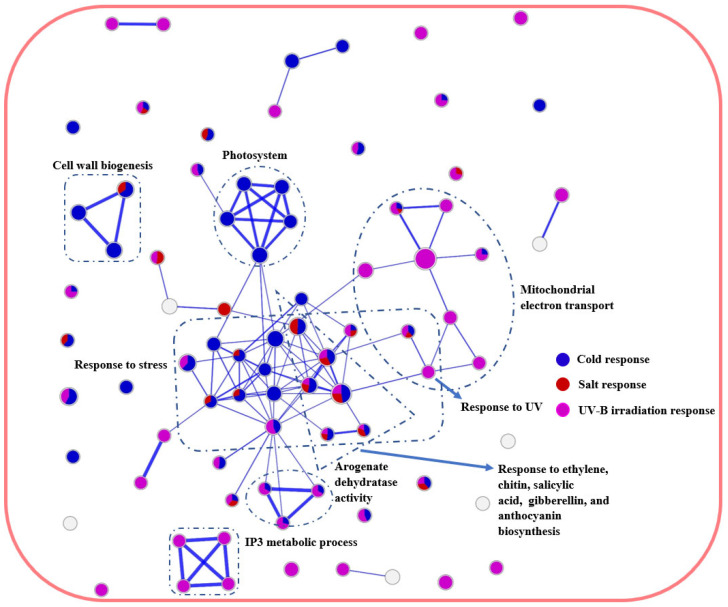
Network analysis of biological processes of P. cheesemanii multiple stress-responsive transcriptomes. Purple, red and blue circles: overrepresented GO terms in response to UV-B radiation, salt and cold stress, respectively. Boxes with dashed lines: clusters of overrepresented GO terms involved in similar biological processes.

**Figure 6 ijms-24-11323-f006:**
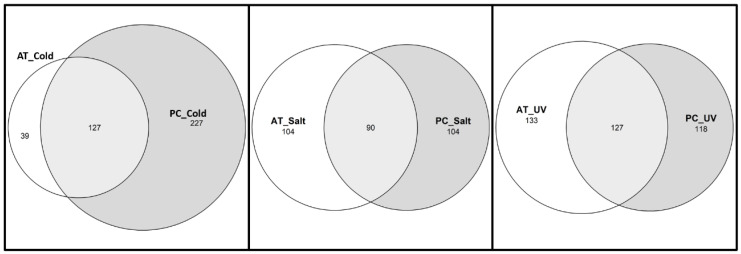
Venn diagrams of the numbers of overrepresented GO terms of *A. thaliana* and *P. cheesemanii* in response to different stresses. Species-specific and overlapping numbers of overrepresented GO terms in response to cold, salt and UV-B radiation stress. AT: *A. thaliana*; PC: *P. cheesemanii*; UV circles: the number of overrepresented GO terms in UV-B radiation (UV) response; Cold circles: the number of overrepresented GO terms in cold stress response; salt circles: the number of overrepresented GO terms in salt stress response.

**Figure 7 ijms-24-11323-f007:**
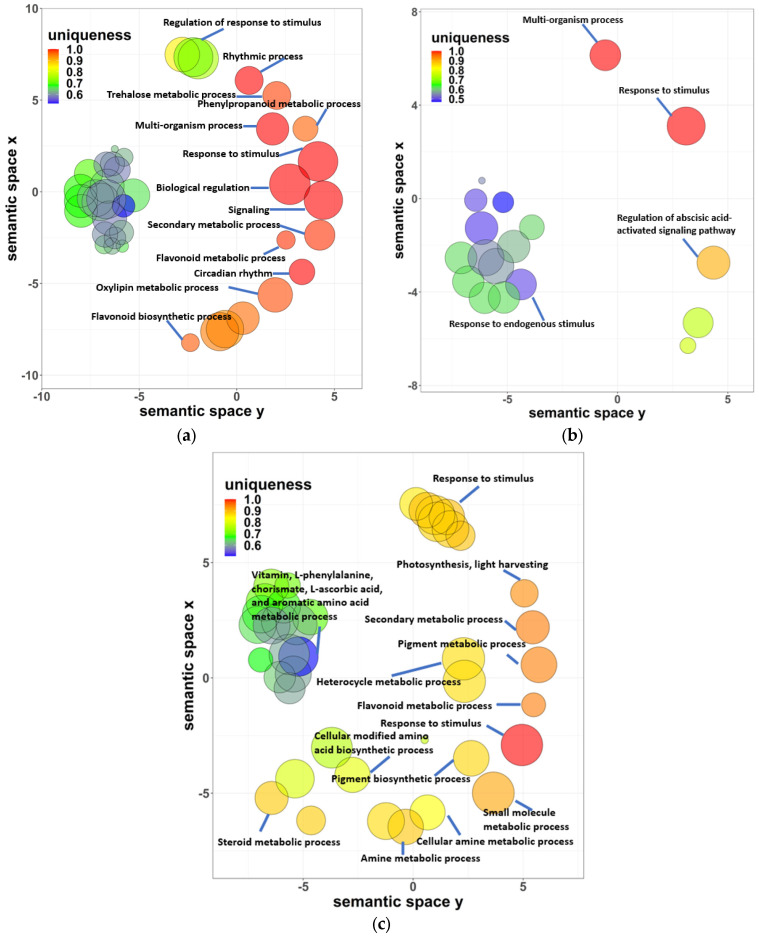
Common GO biological processes of *A. thaliana* and *P. cheesemanii* in response to cold, salt and UV-B radiation stress. Clustering of the common overrepresented terms of GO biological processes of *A. thaliana* and *P. cheesemanii* responding to cold, salt, and UV-B radiation stress. (**a**) Eighty-four common overrepresented GO terms in responding to cold stress were clustered with fourteen representatives of GO biological processes. (**b**) Twenty-six common overrepresented GO terms in responding to salt stress were clustered with four representatives of GO biological processes. (**c**) Sixty-five common overrepresented GO terms in responding to UV-B radiation were clustered with fourteen representatives of GO biological processes. Semantic space X and Y: no intrinsic meaning; uniqueness: measure whether the term is an outlier compared to the list, namely, the negative of average similarity of a term to all other terms. In REVIGO, multi-dimensional scaling was used to reduce the dimensionality of a matrix of the GO terms’ pairwise semantic similarities. First, the terms were placed by using an eigenvalue decomposition of the terms’ pairwise distance matrix. Then, a stress minimization step improved the agreement between the semantic similarities of the terms and their closeness in the two-dimensional space. Thus, the semantically similar GO terms should remain close together in the plot. Figure was generated from REVIGO web (http://revigo.irb.hr/, accessed on 20 October 2021) [30].

**Figure 8 ijms-24-11323-f008:**
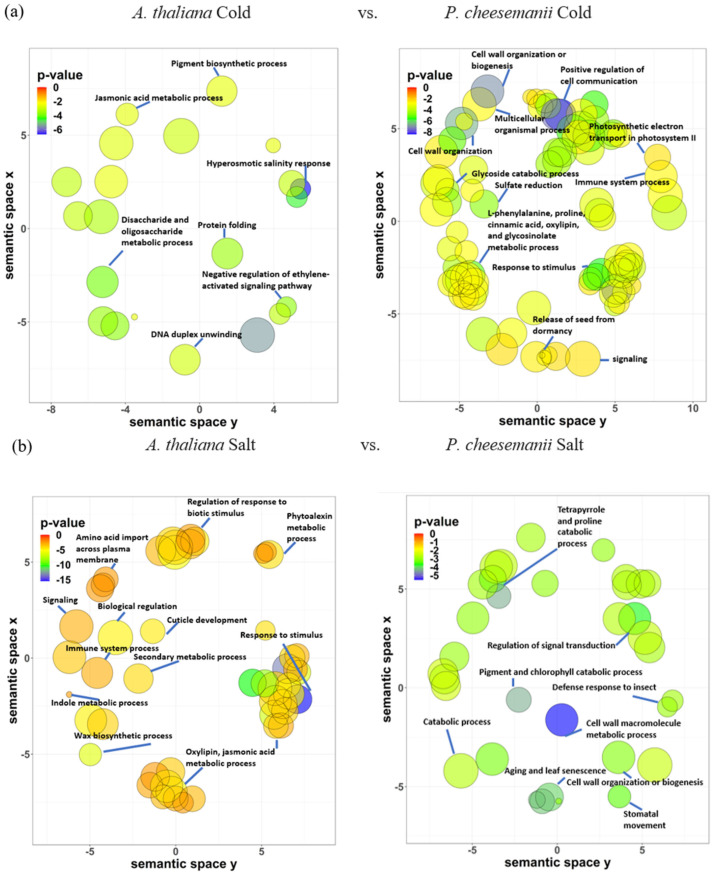
Unique overrepresented terms of GO biological process of *A. thaliana* and *P. cheesemanii* in response to cold, salt and UV-B radiation stress. Clustering of the unique overrepresented terms of GO biological process of *A. thaliana* and *P. cheesemanii* responding to cold, salt and UV-B radiation stress. (**a**) *A. thaliana* Cold vs. *P. cheesemanii* Cold: 32 and 150 unique overrepresented GO terms of *A. thaliana* and *P. cheesemanii* in responding to cold stress were clustered with 7 and 12 representatives of GO biological processes. (**b**) *A. thaliana* Salt vs. *P. cheesemanii* Salt: 90 and 52 unique overrepresented GO terms of *A. thaliana* and *P. cheesemanii* in responding to salt stress were clustered with 11 and 9 representatives of GO biological processes. (**c**) *A. thaliana* UV-B vs. *P. cheesemanii* UV-B: 114 and 57 unique overrepresented GO terms of *A. thaliana* and *P. cheesemanii* in responding to UV-B radiation were clustered with 13 and 5 representatives of GO biological processes. Semantic space X and Y: no intrinsic meaning; uniqueness: measure whether the term is an outlier compared to the list. Namely, the negative of average similarity of a term to all other terms. In REVIGO, multi-dimensional scaling was used to reduce the dimensionality of a matrix of the GO terms’ pairwise semantic similarities. First, the terms were placed by using an eigenvalue decomposition of the terms’ pairwise distance matrix. Then, a stress minimization step improved the agreement between the semantic similarities of the terms and their closeness in the two-dimensional space. Thus, the semantically similar GO terms should remain close together in the plot. Figure was generated from REVIGO web (http://revigo.irb.hr/) [30].

**Figure 9 ijms-24-11323-f009:**
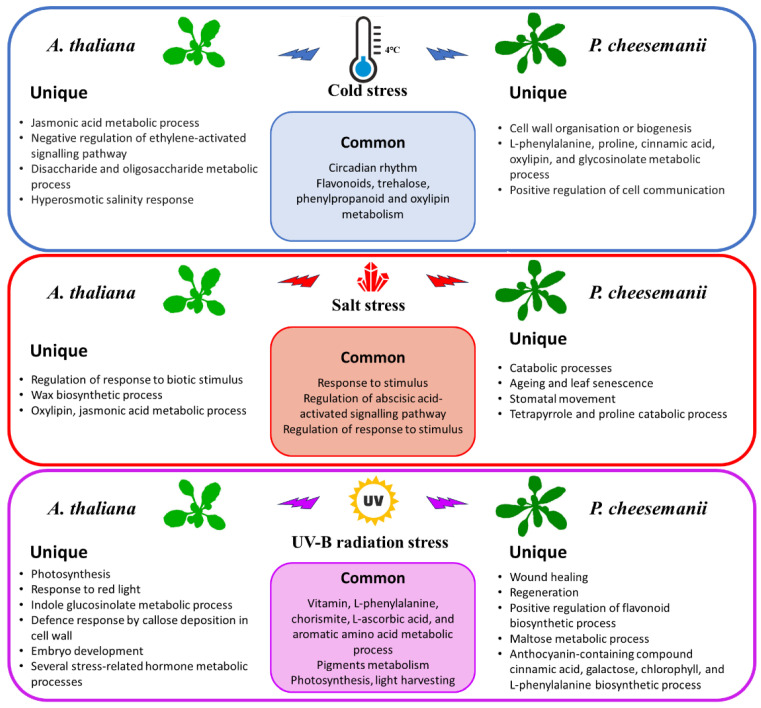
Summary of responses of *A. thaliana* and *P. cheesemanii* plants to cold, salt, and UV-B radiation stresses. Main boxes indicate the three different types of stresses. In each box, main common enrichments in GO terms are indicated in the center, while unique GO enrichments are indicated towards the left for *A. thaliana* and the right for *P. cheesemanii*. See text for details.

**Table 1 ijms-24-11323-t001:** Summary of differentially expressed genes in multiple stresses in *A. thaliana* and *P. cheesemanii*.

Species	Stress	Upregulated Genes	Downregulated Genes	Level I	%	Level II	%	Level III	%	Level IV	%	Total
*A. thaliana*	Cold	2060	1911	2641	66.5	1242	31.3	83	2.1	5	0.1	3971
	Salt	869	185	596	56.5	423	40.1	32	3.0	3	0.3	1054
	UV-B	1181	1033	1763	79.6	404	18.2	43	1.9	4	0.2	2214
*P. cheesemanii*	Cold	2509	1181	2730	74.0	918	24.9	42	1.1	0	0	3690
	Salt	620	186	635	78.8	160	19.9	11	1.4	0	0	806
	UV-B	1005	237	937	75.5	284	22.9	20	1.6	0	0	1242

To group genes based on their expression levels, the following expression categories were defined: Level I, with 2 ≤ |FC| < 4; Level II, with 4 ≤ |FC| < 32; Level III, with 32 ≤ |FC| < 1024 and Level IV, with 1024 ≤ |FC|. FC, fold change.

## Data Availability

Publicly available datasets were analyzed in this study. This data can be found here: the NCBI Sequencing Read Archive (SRA), BioProject ID PRJNA956584.

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
