# Peer review of "Comparative Transcriptomics of Multi-Stress Responses in *Pachycladon cheesemanii* and *Arabidopsis thaliana"

_ijms, 2023, doi:10.3390/ijms241411323_

Round 1

Reviewer 1 Report

The manuscript Comparative transcriptomics of multi-stress responses in Pachycladon cheesemanii and Arabidopsis thaliana conducted comprehensively RNA-seq analysis with two materials. Collectively, the study put forward some interesting opinions, and the manuscript was written carefully and in detail, so, in my opinion, no need major revision for publishing in IJMS. I only have a few questions or suggestions, which possibly need the author to explain or discuss in the manuscript.

1) For the material Pachycladon cheesemanii, I am still not clear for it after reading through the manuscript. How about related studies? Is it more resistant to abiotic stresses? Maybe the author should describe clearly about why to use this two species to conduct this study.

2) For the stress treatment methods, Six-week-old A. thaliana and nine-week-old P. cheesemanii plants were treated, why use different ages? Could using same-age for the two species be better? Treated and control plants were collected after 5 hours of treatment to detect early transcriptional changes. why use 5 hour?

3) The author concentrated on GO throughout the article. How about KEGG analysis?

4) Could the author draw a schematic diagram showing differences between this two species in response to stresses? As after reading throughout the article, we feel jumbly to get the key point for this study.

A minor error: Figure 7. common GO biological process of A. thaliana... should be Common.

Reviewer 2 Report

General Remarks

In manuscript, the authors demonstrate a Comparative transcriptomics of multi-stress responses in Pachycladon cheesemanii and Arabidopsis thaliana.

This study is a worthy attempt to better understand environmental stress of Pachycladon cheesemanii. The topic is timely and will be of interest to the readers of the journal. It definitely deserves to be published and is a valuable contribution to the IJMS journal, Here are some of the suggestions:

Comments

1. Abstract: As a result of this study, specific strategies to alleviate the environmental stress of P. cheesemanii are ambiguous. Specific methods for mitigating environmental stress based on the metabolism involved in glycosinolate, proline and anthocyanin biosynthesis should be proposed.

2. Please rewrite according to the journal format along with resetting the line number.

3. The discussion does not describe differences from previous research results. Genes responding to stress are almost similar or partially different depending on the plant. Based on the results derived from the gene network, GO term, and DEG analysis in this study, the focus should be on why Pachycladon cheesemanii and Arabidopsis thaliana respond differently to environmental stresses.

In the introduction, this phenomenon was described as evolution and adaptation, but in the discussion, it was simply compared with the genes of other crops. Therefore, it seems that the discussion part needs to be rewritten.

Moderate editing of English language required.
